# Asymptotics of transverse momentum broadening in dense QCD media

Paul Caucal⋆

Physics Department, Brookhaven National Laboratory, Upton, NY 11973, USA

⋆ pcaucal@bnl.gov



## Abstract

**We study the asymptotic behaviour of the transverse momentum broadening distribution of an energetic quark or gluon propagating through dense QCD matter, in the large system size $L$ limit, taking into account radiative corrections in the double logarithmic approximation. Thanks to a connection between the evolution of the jet quenching parameter $\hat{q}$ and the formation of traveling wave fronts in nonlinear physics, we obtain a formula for the $L$ dependence of the characteristic transverse momentum scale $Q_s$ of the distribution valid up to terms of order $1/\ln(L)$. We briefly discuss the physical implications of this formula for jet quenching and small-$x$ phenomenology.**

## 1 Introduction

The jet quenching phenomenon, i.e. the strong suppression of high $p_t$ hadrons and jets in ultra relativistic heavy ion collisions at RHIC and the LHC [1,2], is widely regarded as the main signature of the creation of a deconfined QCD matter commonly referred to as the quark-gluon-plasma (QGP) produced in the aftermath of these collisions. The perturbative QCD description of jet quenching relies on the diffusion coefficient $\hat{q}$ which relates the typical transverse momentum square $\langle k_\perp^2 \rangle$ acquired by a highly energetic parton propagating through the plasma with the propagation time $t$ via $\langle k_\perp^2 \rangle \sim \hat{q}t$. In leading order QCD, this linear scaling is a consequence of the random kicks in the plasma, via single gluon exchange with plasma constituents (Coulomb scattering) that leads to an approximate brownian motion in transverse momentum space. These frequent elastic collisions are responsible for interesting emergent phenomena in QCD such as the radiative energy loss via a turbulent gluon cascade [3,4], quantum color decoherence of color single states [5–7] and suppression of phase space for Bremsstrahlung radiations from a highly virtual parton [8].

## 2 Non-linear evolution of the jet quenching parameter

Precision phenomenology of jet quenching leads to the study of the radiative corrections to the coefficient $\hat{q}$. Indeed, gluon emissions also increase the typical transverse momentum square of the energetic parton due to recoil effects. In spite of being suppressed by the strong coupling constant $\alpha_s$, it has been shown that these radiative corrections are enhanced by potentially large double logarithms of the system size $L$, $\sim \alpha_s \ln^2 L$ [4,9–11]. The system size dependence of such contributions is a consequence of the non local nature of quantum corrections.

When $L$ is large, an all-order resummation of these corrections is necessary. In the double logarithmic approximation (DLA), this resummation is achieved thanks to an evolution equation for $\hat{q}(\tau, \boldsymbol{k}_\perp^2)$ ordered in the lifetime $\tau$ of the real or virtual gluon fluctuation, with transverse momentum $\boldsymbol{k}_\perp$ [9–11]:

$$\frac{\partial \hat{q}(Y,\rho)}{\partial Y} = \bar{\alpha}_s \int_{\rho_s(Y)}^{\rho} \mathrm{d}\rho' \, \hat{q}(Y,\rho'), \tag{1}$$

where the variables are defined as follows: $Y = \ln(\tau/\tau_0)$, $\rho = \ln(\boldsymbol{k}_\perp^2/(\hat{q}_0\tau_0))$ and $\bar{\alpha}_s = \alpha_s N_c/\pi$. The infrared cut-off $\tau_0 \ll L$ is a microscopic scale of order of the mean free path [9], while $\hat{q}_0$ is the tree-level value of the diffusion coefficient $\hat{q}$. In principle, the tree-level $\hat{q}$ has a $\boldsymbol{k}_\perp^2$ dependence, but we shall see that the asymptotic limit of $\hat{q}$ for large $Y$ is universal, and does not depend on the details of the tree-level physics. A notable difference of Eq. (1) with respect to the DGLAP evolution equation in the DLA is the presence of the saturation boundary $\rho \gg \rho_s(Y) = \ln(Q_s^2(\tau)/(\hat{q}_0\tau_0))$, i.e. $\boldsymbol{k}_\perp^2 \gg Q_s^2(\tau)$, that enforces the gluon fluctuations to be triggered by a single scattering with plasma constituents. Indeed, the saturation scale $Q_s(\tau)$ is defined as the transverse momentum scale that controls the transition between the multiple soft scattering regime and the single hard scattering one, which leads to the logarithmic enhancement of the gluon fluctuation. This scale is defined by the non-linear relation [12–14]

$$\hat{q}(\tau, Q_s^2(\tau))\tau = Q_s^2(\tau) \Longleftrightarrow \hat{q}(Y, \rho_s(Y)) = \hat{q}_0 \mathrm{e}^{\rho_s(Y)-Y}. \tag{2}$$

In this proceeding, we study the non-linear system of equations (1)-(2) in the large $Y$ limit. The transverse momentum broadening distribution is then related to $\hat{q}(\tau, \boldsymbol{k}_\perp^2)$ by the Fourier transform of the forward scattering amplitude $\mathcal{S}(\boldsymbol{x}_\perp)$ (see e.g. [4,15,16]),

$$\mathcal{P}(\boldsymbol{k}_\perp) = \int \mathrm{d}^2\boldsymbol{x}_\perp \, \mathrm{e}^{-i\boldsymbol{k}_\perp \boldsymbol{x}_\perp} \mathcal{S}(\boldsymbol{x}_\perp), \qquad \mathcal{S}(\boldsymbol{x}_\perp) = \exp\left(-\frac{1}{4}\frac{C_R}{N_c}\hat{q}(\tau_L, 1/\boldsymbol{x}_\perp^2)L\boldsymbol{x}_\perp^2\right). \tag{3}$$

In the scattering amplitude $\mathcal{S}$, $C_R$ is the Casimir factor of the leading parton, and $\tau$ is evaluated along the saturation boundary, i.e. at the time scale $\tau_L$ such $Q_s^2(\tau_L) = 1/\boldsymbol{x}_\perp^2$ if $\tau < L$ and $L$ otherwise [17].

## 3 Scaling limit and sub-asymptotic corrections to $\hat{q}(Y,\rho)$

**Scaling limit.** First, we notice that Eqs. (1)-(2) admit a scaling limit when $Y$ goes to infinity. In mathematical terms, the evolution of $\hat{q}$ is said to possess a scaling limit if there exists a function $f_0$ such that

$$\hat{q}(\tau, \boldsymbol{k}_\perp^2)\tau \underset{\tau\to\infty}{\sim} Q_s^2(\tau) f_0\left(\frac{\boldsymbol{k}_\perp^2}{Q_s^2(\tau)}\right). \tag{4}$$

This scaling property is the analogous of the geometric scaling property of the gluon distribution at small $x$ [18–20]. Assuming that such a scaling limit exists, it is straightforward to get

the function $f_0$ and the leading $Y$ dependence of $Q_s$ from the differential equation (1), with initial condition $f_0(0) = 1$ as a consequence of Eq. (2) [21]:

$$f_0\left(x \equiv \ln\left(\frac{k_\perp^2}{Q_s^2}\right)\right) = e^{\beta x}(1 + \beta x), \qquad \rho_s(Y) = cY + ..., \tag{5}$$

where $\beta = (c-1)/(2c)$ and $c = 1 + 2\sqrt{\bar{\alpha}_s + \bar{\alpha}_s^2} + 2\bar{\alpha}_s$ is the velocity of the traveling wave that propagates to the right on the $\rho$ axis. In the DLA, we can approximate $c \simeq 1 + 2\sqrt{\bar{\alpha}_s}$ and $\beta \simeq \sqrt{\bar{\alpha}_s}$.

**Sub-asymptotic corrections.** To derive the sub-asymptotic corrections to $\hat{q}(Y, \rho)$ and $Q_s(Y)$, we exploit an analogy with the physics of front propagation into unstable states [22], as at stake for instance in the FKPP equation [23, 24]. This is very similar to the calculation of the asymptotic expansion of the saturation scale and solutions to the BK equation [25, 26] in the small-$x$ limit [27–29]. Following the method of Ebert and Van Saarloos [30], we consider two distincts expansions around the scaling limit called "front interior expansion" and "leading edge expansion":

$$\text{Front interior: } \hat{q}(Y, \rho) = \hat{q}_0 e^{\rho_s(Y)-Y} e^{\beta x}\left(e^{-\beta x} f_0(x) + \frac{1}{Y^\alpha} f_1(x) + ... + \frac{1}{Y^{n\alpha}} f_n(x) + ...\right), \tag{6}$$

$$\text{Leading edge: } \hat{q}(Y, \rho) = \hat{q}_0 e^{\rho_s(Y)-Y} e^{\beta x}\left(Y^\alpha F_\alpha\left(\frac{x}{Y^\alpha}\right) + F_0\left(\frac{x}{Y^\alpha}\right) + Y^{-\alpha} F_{-\alpha}\left(\frac{x}{Y^\alpha}\right) + ...\right), \tag{7}$$

for a power $\alpha > 0$ to be determined. As in the traveling waves solutions to the FKPP problem [31], the front interior expansion describes the behaviour of the front in its rest frame ($x$ fixed), whereas the leading edge expansion focuses on the regime where $x \sim Y^\alpha$, which is large when $Y \to \infty$. The matching of these two expansions enables one to extract the asymptotic expansion of the velocity of the pulled front:

$$\frac{d\rho_s(Y)}{dY} = c + \frac{b_1}{Y} + \frac{b_2}{Y^{3/2}} + ..., \tag{8}$$

beyond the first term provided by the scaling limit. Plugging these two expansions inside Eq. (1), one obtains a triangular system of differential equations for $f_n$ and $F_m$ respectively. The homogeneity of the equation satisfied by $F_\alpha$ gives $\alpha = 1/2$ [21]. The non-linear equation (2) provides the initial conditions that fix all the constants of integration. On the other hand, the boundary conditions at $z = x/Y^\alpha \to \infty$ determines the constants [1]

$$b_1 = -\frac{3c}{1+c}, \qquad b_2 = \frac{3c\sqrt{2\pi(c-1)}}{(1+c)^2}. \tag{9}$$

We find that the functions $f_n(x)$ are all polynomials in $x$, with leading powers in $x$ giving the analytic series of $F_{1/2}$, the sub-leading powers giving the analytic series of $F_0$ and so on. More concretely, the first two terms read [21]

$$f_1(x) = 0, \qquad f_2(x) = \frac{b_1 x}{c^2}\left[1 + \frac{(c-1)(3+c)}{8c}x + \frac{(c-1)^2(1+c)x^2}{48c^2}\right], \tag{10}$$

while the leading edge expansion reads [21]

$$F_{1/2}(z) = \beta z \exp\left(-\frac{\beta z^2}{4c}\right), \qquad F_0(z) = \exp\left(-\frac{\beta z^2}{4c},\right) h(z^2), \tag{11}$$

---

[1]The details of this calculation will be reported and discussed elsewhere [32].

with $h$ solution of $32c^4 u h'' + 4c^2(4c^2 + (1-c)u)h' - 4c^2(1-c)h = b_2(1-c)^2(1+c)\sqrt{u}$ and initial conditions $h(u) = 1 + \mathcal{O}(u)$.

Finally, we obtain the following universal behaviour of the saturation scale that controls the transverse momentum distribution:[2]

$$\rho_s(Y) = cY - \frac{3c}{1+c}\ln(Y) - \frac{6c\sqrt{2\pi(c-1)}}{(1+c)^2}\frac{1}{\sqrt{Y}} + \mathcal{O}\left(\frac{1}{Y}\right), \tag{12}$$

which is the main result of this proceeding. We emphasize that this result is different from the one obtained in [33, 34] for which the lower boundary of the $\rho'$ integral in (1) is set to $Y$ instead of $\rho_s(Y)$. The difference is parametrically of order $\sqrt{\alpha_s}\ln\ln L$ [21], and therefore dominant over single-logarithmic corrections that we do not consider in this analysis.

## 4 Discussion and conclusion

The scaling limit of the quenching parameter $\hat{q}$ and the resulting transverse momentum distribution has a nice physical interpretation in terms of a special kind of random walk followed by the leading parton in transverse momentum space known as Levy flight [21]. Levy flight leads notably to a super-diffusive behaviour [35]: the typical width of the transverse momentum distribution scales like $\langle k_\perp^2 \rangle \propto L^c$. Since $c > 1$, this scale grows with time faster than standard (Brownian) diffusion, as a result of the non-linearity and self-similarity of multiple gluon radiations. On the other hand, the transverse momentum broadening distribution exhibits power law tail at large $k_\perp$, with a weaker power than the Rutherford one $1/k_\perp^4$, characteristic of point-like interactions [21]:

$$\mathcal{P}(k_\perp) \underset{k_\perp^2 \gg Q_s^2}{\propto} \frac{1}{k_\perp^{4-2\beta}}. \tag{13}$$

This heavy tail is also characteristic of the probability density for the position of a particle undergoing a Lévy flight process in two dimensions.

Remarkably, the asymptotic expansion (12) is universal [21], as well as the form of the sub-asymptotic corrections provided by the functions $f_0$, $f_2$, $F_{1/2}$ and $F_0$. It means that these results are independent of the tree-level initial condition for the transverse momentum broadening distribution. In the context of small-$x$ physics, it can therefore provide a pQCD motivated functional form for the initial conditions [36, 37] to the BK equation, that includes gluon fluctuations enhanced by double logs of the nucleus target size to all orders.

Finally, we point out that even though the expression (12) is an asymptotic expansion, it provides a good approximation of the saturation scale down to small values of $L$, close to those relevant in heavy-ion phenomenology, thanks to the determination of the sub-leading terms up to corrections of order $1/\ln(L)$. Nevertheless, precise phenomenology requires to go beyond the double logarithmic approximation, including at least running coupling effects. We leave this calculation for an upcoming publication [32].

**Funding information** This work was supported by the U.S. Department of Energy, Office of Science, Office of Nuclear Physics, under contract No. DE- SC0012704.

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
