# Peer review of "Asymptotics of transverse momentum broadening in dense QCD media"

_SciPost Physics Proceedings, doi:SciPost Phys. Proc. 10, 018 (2022)_

## Round 1 · Referee Report · Anonymous · 2022-2-16

Strengths

Good context-setting in terms of jet quenching.

Mathematical content well explained, though relying on some expert knowledge of context. Reasonable within space constraints.

Good discussion of the interpretation of the results in terms of clearly explained Levy flights.

Weaknesses

It would have been nice in the conclusions to go a little further in explaining the consequences of the Levy-flight behaviour in terms of "quenching power", i.e. if the <kT^2> distribution grows faster than from Brownian motion, does this imply a greater than naively expected broadening/quenching from QGP than from a "normal" medium? And is there a physical intuition for the super-diffusive behaviour, e.g. in terms of pQCD-correlated emissions?

A few minor spelling/grammar typos, below threshold for correction in proceedings.

Report

A clearly written submission, covering a technical area with material accessible to generalists as well as specialists. Well suited for publication without revision.

---

## Editorial Decision

published